# Overexpression and Biochemical Characterization of an Endo-α-1,4-polygalacturonase from *Aspergillus nidulans* in *Pichia pastoris*

**DOI:** 10.3390/ijms21062100

**Published:** 2020-03-19

**Authors:** Hua Xu, Pengfei Zhang, Yuchen Zhang, Zebin Liu, Xuebing Zhang, Zhimin Li, Jian-Jun Li, Yuguang Du

**Affiliations:** 1Jiangxi Agricultural University, College of Bioscience and Bioengineering, Nanchang 330045, China; 15756291121@163.com; 2Sichuan Normal University, College of Life Science, Chengdu 610101, China; 18091765127@163.com; 3National Key Laboratory of Biochemical Engineering, National Engineering Research Center for Biotechnology (Beijing), Key Laboratory of Biopharmaceutical Production & Formulation Engineering, PLA, Institute of Process Engineering, Chinese Academy of Sciences, Beijing 100190, China; yczhang@ipe.ac.cn (Y.Z.); xbzhang@ipe.ac.cn (X.Z.); ygdu@ipe.ac.cn (Y.D.); 4Capital Normal University, College of Life Sciences, Beijing 10048, China; zebinl@163.com

**Keywords:** endo-α-1,4-polygalacturonase, *Aspergillus nidulans*, pH and temperature-rate profile, pH and thermal stability, identification of hydrolysis products, structure modelling

## Abstract

Pectinases have many applications in the industry of food, paper, and textiles, therefore finding novel polygalacturonases is required. Multiple sequence alignment and phylogenetic analysis of AnEPG (an endo-α-1,4-polygalacturonase from *Aspergillus nidulans*) and other GH 28 endo-polygalacturonases suggested that AnEPG is different from others. AnEPG overexpressed in *Pichia pastoris* was characterized. AnEPG showed the highest activity at pH 4.0, and exhibited moderate activity over a narrow pH range (pH 2.0–5.0) and superior stability in a wide pH range (pH 2.0–12.0). It displayed the highest activity at 60 °C, and retained >42.2% of maximum activity between 20 and 80 °C. It was stable below 40 °C and lost activity very quickly above 50 °C. Its apparent kinetic parameters against PGA (polygalacturonic acid) were determined, with the *K_m_* and *k_cat_* values of 8.3 mg/mL and 5640 μmol/min/mg, respectively. Ba^2+^ and Ni^2+^ enhanced activity by 12.2% and 9.4%, respectively, while Ca^2+^, Cu^2+^, and Mn^2+^ inhibited activity by 14.8%, 12.8%, and 10.2% separately. Analysis of hydrolysis products by AnEPG proved that AnEPG belongs to an endo-polygalacturonase. Modelled structure of AnEPG by I-TASSER showed structural characteristics of endo-polygalacturonases. This pectinase has great potential to be used in food industry and as feed additives.

## 1. Introduction

Pectin is one of the most important components in the middle lamella and cell wall of plants, and accounts for one-third of the dry weight of plant material [1,2,3]. It plays multiple functions during plant growth, including morphogenesis, defense, cell adhesion, cell wall structure, cellular expansion, and so on [2,4]. As one of the most complex polysaccharides in plant cell walls, pectin exists in a variety of structures such as homogalacturonan (HG), xylogalacturonan, rhamnogalacturonan I and rhamnogalacturonan II, among which HG is the most abundant (<65%) [2,3].

In nature, pectin is degraded by pectin-degrading enzymes [5,6,7,8,9]. Due to complexity and heterogeneousity, complete degradation of pectin requires the combined action of esterases, lyases, and hydrolases [5,6,7,8,9]. Esterases mainly include pectin methylesterases (EC 3.1.1.11), acetylesterases (EC 3.1.1.6), and feruloyl esterases (EC 3.1.1.73), which remove methyl, acetyl, and feruloyl of pectin, respectively. Pectate lyases (EC 4.2.2.2) and pectin lyases (EC 4.2.2.10) catalyze the cleavage of the α-1,4 glycosidic bond of pectin and pectate by transeliminative reactions, respectively, to produce Δ4,5 unsaturated products. Hydrolases cleave the α-1,4-linkage of pectin and pectate, including endo-polygalacturonases (EC 3.2.1.15), exo-polygalacturonases (EC 3.2.1.67), rhamnogalacturonases (EC 3.2.1.171), etc. [5,6,7,8,9]. Among the pectinolytic enzymes, endo-polygalacturonases (endo-PGs) are most extensively investigated, and are classified into family GH 28 of CAZy (carbohydrate-active enzymes database) based on sequence and structure similarity [5,6,7,8,9]. Most characterized endo-PGs are from fungi, such as the species of *Aspergillus* and *Penicillium*. For example, some endo-PGs were purified and characterized from different genera of *Aspergillus*, including *A. niger* [10,11,12], *A. awamori* [13], *A. carbonarius* [14], *A. aculeatus* [15], and *A. flavus* [16]. Some endo-PGs from different species of *Penicillium* were also investigated, for example, *Penicillium oxalicum* CZ1028 [17], *Penicillium* sp. CGMCC 1669 [18], and *Penicillium occitanis* [19]. Many endo-PGs from other fungi species were characterized as well, for example, *Neosartorya fischeri* P1 [20], *Bispora* sp. MEY-1 [21], *Achaetomium* sp. Xz8 [22], *Talaromyces leycettanus* JCM 12802 [23], and *Thielavia arenaria* XZ7 [24].

Pectinases have found many applications in the industry of food, paper and pulp, and textiles [5,6,7,8,9]. Considering the fact that pectinases account for a considerable proportion of enzyme markets, efforts to find new polygalacturonases with good properties are needed. *Aspergillus nidulans* is one species of filamentous fungi in the phylum Ascomycota, which grows mainly on dead or decaying plant material. The *A. nidulans* genome was sequenced in 2005 [25], and the putative plant cell wall polysaccharide degrading enzymes were assigned to 166 ORFs (open reading frames) [26]. A later study in which 72 ORFs of *A. nidulans* were expressed in *Pichia pastoris* confirmed the predicted functions for them [27,28]. Thus, *A. nidulans* has become a good source for mining novel polysaccharide-degrading enzymes. Some polysaccharide-degrading enzymes from *A. nidulans* were identified and characterized [28,29]. However, no endo-PGs from *A. nidulans* were investigated functionally and in detail. Multiple sequence alignment and phylogenetic analysis of AnEPG (an endo-α-1,4-polygalacturonase from *A. nidulans*, GeneBank accession number AN8327.2) and other GH 28 endo-polygalacturonases implied that AnEPG is different from others [15,17,22,24,30,31] (Appendix A). In this study, AnEPG was overproduced in *P. pastoris*, and biochemically characterized in detail for the first time. Moreover, characterizing the hydrolysis products and modelling the structure of this enzyme were carried out.

## 2. Results and Discussion

### 2.1. Sequence Alignment and Phylogenetic Analysis of AnEPG with Other GH 28 endo-PGs

Though many polysaccharide-degrading enzymes from *A. nidulans* were characterized, so far no endo-PGs from this fungus have been characterized in detail [28,29]. According to the genome sequence of *Aspergillus nidulans* FGSC A4, the gene (Gene ID: 2868744) encoding a hypothetical protein (GenBank accession number AN8327.2) belonging to the GH 28 family, was named as AnEPG (endo-α-1,4-polygalacturonase from *Aspergillus nidulans*) in the current study. A 19-residue signal peptide with a putative processing site (VMA-TP) was identified using the SignalP 4.1 server. The potential *O*- and *N*-glycosylation sites were predicted to be Thr27, Thr28, Ser30, Thr32, Ser38, and Asn154, Asn192, Asn371, respectively.

Homologous GH 28 endo-PGs were found by subjecting them to a BLAST search of the sequence of AnEPG. Multiple sequence alignment and the phylogenetic analysis of AnEPG and other GH 28 endo-PGs revealed that AnEPG is different from other endo-polygalacturonases (Appendix A). AnEPG exhibited the highest sequence identity to a hypothetical protein from *Aspergillus mulundensis* (GenBank accession no ABL01533 XP026600476.1) (91.8%), and 76.25% sequence identity to EPG4 from *Penicillium oxalicum* CZ108 [17] (Appendix A). Four putative disulfide bonds are formed by Cys39 and Cys57, Cys217, and Cys233, Cys345, and Cys350, Cys369 and Cys378, three of which (Cys39-Cys57, Cys217-Cys233, Cys345-Cys350) are highly conserved among GH 28 endo-PGs (Appendix A).

Structural models of AnEPG based on homologous enzymes were obtained by the I-TASSER server [32]. Five top ranking 3D models were generated. Each model was validated based on C-score (confidence score), TM-score (template modeling score), RMSD (the root-mean-square deviation), and cluster density. In general, models with C-score > −1.5 have a correct fold [32]. Model 1 had the highest C-score (1.69) value reflecting a model of better quality (TM-score = 0.95 ± 0.05 and RMSD = 3.1 ± 2.2 Å) (Figure 1). Similarly to homologous GH28 endo-PGs [30,31,33], the predicted three-dimensional structure of AnEPG was a right handed parallel β-helix with 12 (10 complete) turns, in which the β-strands were separated by turns that consisted of either a sharp bend or a loop (Figure 1). Based on structural and sequence alignment of AnEPG and AaEPG from *Aspergillus aculeatus* and pga II from *Aspergillus niger*, residues Asp194, Asp215, Asp216, and His237 of AnEPG, which are the equivalent of Asp159, Asp180, Asp181, and His202 of AaEPG, and Asp180, Asp201, Asp202, and His223 of pga II, were predicted to be involved in catalysis (Figure 1) (Appendix A) [30,33].

### 2.2. Overexpression of AnEPG in P. pastoris

AnEPG overexpressed in *P. pastoris* X-33 was first produced in flasks. AnEPG was induced with 0.5% methanol. The protein expression level increased with the induction time, and 96-h induction gave the highest protein expression yield (Appendix A). A band corresponding to 75 kDa was observed, which was much higher than the predicted molecular weight of AnEPG (around 42 kDa) (Appendix A). The difference between the predicted molecular weight and the apparent one on SDS PAGE for AnEPG was possibly due to the fact that highly glycosylated proteins are usually obtained when they are overexpressed in *P. pastoris* [34]. The identity of AnEPG was also confirmed by enzymatic hydrolysis of PGA (polygalacturonic acid) with specific activity of 3268.6 U/mg. AnEPG exhibited much higher specific activity against PGA than endo-PG I from *Penicillium* sp. CGMCC 1669 (815.5 U/mg) [18], endo-PGA1 from *Bispora* sp. MEY-1 (1520 U/mg) [21], and AaEPG from *Aspergillus aculeatus* (1892 U/mg) [15], and greatly lower than Nfpg II from *Neosartorya fischeri* P1 (11,793 U/mg) [20], endo-TePG28b from *Talaromyces leycettanus* (25,900 U/mg) [23], PG I from *Achaetomium* sp. Xz8 (28,122 U/mg) [22], PG2 from *Penicillium occitanis* (31,397.3 U/mg) [19], endo- PG7fn from *Thielavia arenaria* XZ7 (34,382 U/mg) [24], and Nfpg I from *Neosartorya fischeri* P1 (40,123 U/mg) [20].

### 2.3. Determination of pH Optima and pH Stability of AnEPG

As shown in Figure 2, AnEPG showed the highest activity at pH 4.0, and no obvious activities were detected above pH 6.0. The enzyme was active in a narrow pH range (pH 2.0–5.0), and retained >54.1% of maximum activity between this pH range. Thus, AnEPG was classified as acidic endocellulases. It appears that many endo-polygalacturonases belonged to acidic ones with optimal pH values around 3.5–6.0, including endo-TePG28b (pH 3.5) [23], endo-PGA1 (pH 3.5) [21], endo-PG I (pH 3.5) [18], Nfpg II (pH 4.0) [20], PGA-ZJ5A from *Aspergillus niger* ZJ5 (pH 4.5) [12], PG7fn (pH 5.0) [24], AaEPG (pH 5.0) [15], Nfpg I (pH 5.0) [20], endo-PG I (pH 6.0) [22], and PG2 (pH 6.0) [19]. Only several endo-polygalacturonases were classified as alkaline, such as Endo-PG from *Aspergillus fumigatus* MTCC 2584 with optimal pH 10.0 [35].

The pH stability of AnEPG was also investigated (Figure 3). Notably, AnEPG was stable between pH 2.0 and pH 12.0, retaining more than 62.7% of original activity after 120 h. In comparison with other endo-polygalacturonases, it showed superior stability over a wide pH range, whereas others exhibited high stability only over a narrow pH range, such as Nfpg I (pH 5.0–7.0) [20], AaEPG (pH 2.0–6.0) [15], PGA-ZJ5A (pH 2.0–6.0) [12], Nfpg II (pH 2.0–6.0) [20], endo-PG I (pH 2.0–6.0) [18], endo-PG I (pH 3.5–8.0) [22], endo-TePG28b (pH 2.0–7.0) [23], endo-PGA1 (pH 2.0–7.0) [21], PG7fn (pH 3.0–8.0 ) [24], and PG2 (pH 4.0–9.0) [19].

### 2.4. Determination of Optimal Temperature and Thermal Stability of AnEPG

The optimal temperature of AnEPG was determined to be 60 °C, and retained > 51.0% of maximum activity between 30 and 80 °C. At 20 °C, it maintained 42.2% of maximum activity. Most characterized endo-polygalacturonases belonged to mesophilic ones with optimal activities at around 35–50 °C, including PG2 (35 °C) [19], endo-PG I (40 °C) [18], PGA-ZJ5A (40 °C) [12], endo-PG I (45 °C) [22], and AaEPG (50 °C) [15]. In contrast, some other endo-polygalacturonases, like AnEPG, exhibited optimal activities at ≥55 °C, for example, endo-PGA1 (55 °C) [21], AnEPG (60 °C), PG7fn (60 °C) [24], Nfpg II (65 °C) [20], and endo-TePG28b (70 °C) [23].

The thermal stability of AnEPG was studied after being pre-incubated for a fixed time at pH 6.0, and at 30, 40, and 50 °C, respectively (Figure 4). AnEPG was stable at 30 °C, and lost only 11.5% of original activity after 120 min. It retained 80.7% of initial activity at 40 °C after 120 min. However, it was completely inactivated at 50 °C after 120 min, and retained only 39.1% of original activity at 50 °C after 15 min. Therefore, AnEPG was thermally stable to some extent. Just like AnEPG, many endo-polygalacturonases were stable at ≤40 °C, such as PG2 (≤35 °C) [19], endo-PG I (≤40 °C) [18], PGA-ZJ5A (≤40 °C) [12], AaEPG (≤40 °C) [15], and endo-PG I (≤45 °C) [22]. In comparison, some endo-polygalacturonases, including PG7fn (≤50 °C) [24], endo-PGA1 (≤55 °C) [21], endo-TePG28b (≤60 °C) [23], and Nfpg II (≤60 °C) [20], exhibited superior thermal stability at ≥50 °C.

### 2.5. Determination of Kinetic Parameters of AnEPG

The kinetic parameters of recombinant AnEPG against CMC were determined. Since saturation was not achieved even when high PGA concentrations were used, the deduced kinetic parameters were apparent (Figure 5). The apparent *K_m_* and *V_max_* values of AnEPG towards PGSA were 8.3 ± 2.2 mg/mL and 5640 ± 300 μmol/min/mg, respectively. 

In comparison with the kinetic parameters of other endo-polygalacturonases, the apparent *K_m_* value of AnEPG is much lower than that of endo-PG I (19.5 mg/mL) and AaEPG (15.1 mg/mL), showing higher binding affinity towards PGA than endo-PG I and AaEPG. However, its apparent *K_m_* value is significantly higher than that of other endo-polygalacturonases such as endo-PG I (0.32 mg/mL), Nfpg II (0.5 mg/mL), PGA-ZJ5A (0.85 mg/mL), endo-TePG28b (1.2 mg/mL), endo-PGA1 (1.25 mg/mL), and PG7fn (2.0 mg/mL), demonstrating lower binding affinity of AnEPG towards PGA than these endo-polygalacturonases. Though AnEPG showed a much higher apparent *V_ma_*_x_ value than endo-PGA1 (1.25 mg/mL, 2526 U/min/mg), endo-PG I (909.1 U/min/mg), and PGA-ZJ5A (1.87 μmol/min/mg), its apparent *V_max_* value was significantly lower than that of endo-PG I (97,951 μmol/min/mg), endo-TePG28b (63,694 μmol/min/mg), PG7fn (32,000 μmol/min/mg), and Nfpg II (15,053 μmol/min/mg).

According to the modelled structure of the AaEPG-octagalacturonate complex [33], 17 residues, which are possibly involved in interaction with substrate, were identified, including Lys105, His110, Gln128, Asp150, His156, Asn157, Asp162, Asp180, Asn186, Ser208, His202, Arg212, Asp231, Arg235, Lys237, Lys61, Tyr270. Based on sequence and structural alignment of AaEPG (PDB ID: 1IA5) and modelled AnEPG, 15 residues are completely conserved, and only two residues Asp231 and Lys261 of AaEPG, corresponding to Val266 and Asp296 of AnEPG, are not conserved. The difference between these two residues might lead to different *K_m_* values of AnEPG and AaEPG, which needs further investigation.

### 2.6. Effects of Divalent Metal Ions on Enzyme Activity 

The effects of divalent metal ions on AnEPG activity were examined (Figure 6). Ba^2+^ and Ni^2+^ upregulated the pectinolytic activity of AnEPG by 12.2% and 9.4%, respectively, while Ca^2+^, Cu^2+^ and Mn^2+^ decreased the activity of AnEPG by 14.8%, 12.8%, and 10.2% separately. Other divalent metal ions did not show an obvious influence on the catalytic activity of AnEPG.

Combining our results with the published ones, it seems that some divalent metal ions had a different impact on endo-polygalacturonases from different microorganisms (Appendix A). For instance, Ni^2+^ interfered with the activities of endo-PG I [18], Nfpg II [20], and PGA-ZJ5A [12], and did not have any effects on endo-PGA1 [21], PG7fn [24], and endo-PG I [22]. However, it enhanced the activity of AnEPG. Ba^2+^ showed an inhibitory impact on PG2 [19], whereas it upregulated the activity of AnEPG. Ca^2+^ reduced the activities of endo-PGA1 [21], endo-PG I [18], Nfpg II [20], PGA-ZJ5A [12], PG7fn [24], endo-PG I [22], and AnEPG, while it did not impact AaEPG [15] and PG2 [19]. Cu^2+^ inhibited endo-PGA1 [21], endo-PG I [18], Nfpg II [20], endo-PG I [22], PG2 [19], and AnEPG, and it did not exhibit any influence on PG7fn [24]. However, Cu^2+^ activated AaEPG [15]. Mn^2+^ acted as an inhibitor of endo-PG I [18], PGA-ZJ5A [12], AaEPG [15], PG2 [19], and AnEPG, and as an activator of endo-PGA1 [21], and did not impact Nfpg II [20] and PG7fn [24]. Zn^2+^ displayed an inhibitory effect on endo-PGA1 [21], endo-PG I [18], Nfpg II [20], PGA-ZJ5A [12], and AaEPG [15], while it did not influence PG7fn [24], endo-PG I [22], and AnEPG. Co^2+^ inhibited endo-PG I [18], PGA-ZJ5A [12], and PG2 [19], and activated Nfpg II [20], and did not affect endo-PGA1 [21], PG7fn [24], endo-PG I [22], and AnEPG. Mg^2+^ acted as an inhibitor of Nfpg II [20], PGA-ZJ5A [12], and endo-PG I [22], and acted as an activator of PG [19], and did not show any influence on endo-PGA1 [21], endo-PG I [18], PG7fn [24], AaEPG [15], and AnEPG. Fe^2+^ displayed a stimulatory effect on AaEPG [15], whereas it inhibited PG2 [19]. By contrast, it did not influence PGA-ZJ5A [12] and AnEPG.

### 2.7. Analysis of Hydrolysis Products of PGA by AnEPG

TLC (thin-layer chromatography) analysis of the soluble sugars released from PGA by AnEPG indicated that digalacturonate and other oligogalacturonates were produced at the initial stage (20 min) and accumulated as hydrolysis continued (Figure 7). The monomer galacturonate was observed after 3 h. The presence of galacturonate released from oligogalacturonates by AnEPG suggested that it was a typical endo-acting enzyme, which preferentially cleaved the internal glycosidic bonds of oligogalacturonates and pectin/pectate.

## 3. Materials and Methods

### 3.1. Materials 

Chemicals were from Sigma, Merck or Ameresco. *Pichia pastoris* X-33 overexpressing the endo-α-1,4-endopolygalacturonase gene (GeneBank accession number AN8327.2) from *Aspergillus nidulans* was purchased from FGSC (Fungal Genetics Stock Center, Manhattan, KS, USA).

### 3.2. Bacterial Strains, Plasmids, and Media

*P. pastoris* X-33 was grown in YPD (yeast extract peptone dextrose) medium (1% yeast extract, 2% peptone, and 2% glucose) at 30 °C or on YPD supplemented with 1.5% (*w*/*v*) agar. For AnEPG overexpression, *P. pastoris* X-33 was first grown overnight in BMGY (buffered complex glycerol medium) (1% yeast extract, 2% peptone, 1% glycerol, 1.34% YNB, 4 × 10^-5^ g/L biotin, and 0.1 M potassium phosphate buffer, pH 6.0), then in BMMY (buffered complex methanol medium) (1% yeast extract, 2% peptone, 1% methanol, 1.34 % YNB, 4 × 10^-5^ g/L biotin, and 0.1 M potassium phosphate buffer, pH 6.0) for a couple of days.

### 3.3. Protein Overexpression

Protein expression was induced with 0.5% (*v*/*v*) methanol in baffled flasks (100 ml BMMY) for four days. The supernatants were precipitated with 80% (NH_4_)_2_SO_4_, and the precipitated proteins were redissolved in buffer A (50 mM Tris/HCl, pH 8.0, 0.5 M NaCl). The protein concentration was determined by the Bradford method using bovine serum albumin as a standard. 

### 3.4. Enzyme Activity Assay

All enzyme assays were done in triplicate. Endo-α-1,4-polygalacturonase activity was determined by measuring the amount of reducing sugars released from PGA (polygalacturonic acid) through the DNS (3,5-dinitrosalicylic acid) method [36]. D-(+)-galacturonic acid was used as a standard. Enzymatic reactions were performed in the presence of 0.5% PGA (*w*/*v*) in 50 mM B & R (Britton and Robinson) buffer at 37 °C for 15 min. One unit (U) of endo-α-1,4-polygalacturonase activity toward PGA was defined as the amount of protein required to release 1 μmol of reducing sugar per min under standard assay conditions, and specific activity was defined as units mg^-1^ protein.

### 3.5. Determination of Optimal pH and pH Stability

The optimal pH of AnEPG was determined in 50 mM B & R buffer at 37 °C and pH between 2.0 and 12.0, and all enzymatic reactions were incubated for 15 min. The pH stability was estimated by first preincubating PGA in 50 mM B&R buffer at different pH values (pH 2.0 – 12.0) at 4 °C for 2, 24, and 120 h respectively. The residual activities were then determined at 37 °C and optimal pH, and the percentage of the residual activity at different time points and pH values against the initial one was calculated.

### 3.6. Determination of Optimal Temperature and Thermal Stability

The optimal temperature of AnEPG was determined in 50 mM B & R buffer (pH 4.0) between 20 and 80 °C, and all enzymatic reactions were incubated for 15 min. To determine the thermal stability of AnEPG, it was pre-incubated for varied time intervals (15 min to 2 h) at pH 6.0, and 30, 40, and 50 °C, respectively. The residual activities were measured at 37 °C and optimal pH, and the percentage of the residual activity at different time points and temperatures against the original one was calculated.

### 3.7. Determination of Kinetic Parameters

Endo-α-1,4-polygalacturonase activity was measured at 37 °C using PGA as substrate at concentrations ranging from 0.2% to 2% (*w*/*v*) in 50 mM B & R buffer (optimal pH). The release of reducing sugars was quantified after being incubated for 5 min, and kinetic parameters were determined based on the Michaelis–Menten equation.

### 3.8. Effects of Divalent Metal Ions on Enzyme Activity

The effects of divalent metal ions on the catalytic activity of AnEPG were determined in the presence of various divalent metals (Pb(CH_3_COO)_2_, NiSO_4_, MnSO_4_, CuSO_4_, BaCl_2_, ZnSO_4_, CoCl_2_, CaCl_2_, MgCl_2_, and FeSO_4_). Since phosphate in B & R buffer may interfere with the enzyme assay, 100 mM sodium acetate (optimal pH) was used. The percentage of the activity in the presence of different divalent metal ions against the control without metal ions was calculated.

### 3.9. Thin Layer Chromatography Analysis of Hydrolysis Products of PGA by AnEPG

To demonstrate the mode of action of AnEPG, hydrolysis of PGA (0.5%) by AnEPG (0.89 μg/mL) was carried out in 50 mM B & R buffer (optimal pH) at 37 °C for 5 min, 10 min, 20 min, 30 min, 1 h, 2 h, 3 h, and 24 h, respectively, and a control in the absence of AnEPG was also set up. The hydrolysis products of PGA by AnEPG were analyzed by TLC (thin-layer chromatography). Samples were spotted on the silica gel plate and the TLC plate was placed in the mixture (butan-1-ol:acetic acid:water (9:4:7 (*v*/*v*) as the mobile phase [19]. At the end of migration, products were visualized by heating at 105°C for 5 min after spraying the plates with 10% sulfuric acid.

### 3.10. Sequence Analysis

The homologous sequences were searched using the BLAST online server (http://blast.ncbi.nlm.nih.gov/Blast.cgi). Multiple sequence alignment was performed by Clustal Omega (https://www.ebi.ac.uk/Tools/msa/clustalo/). A phylogenetic tree was constructed from multiple alignment of AnEPG and other GH 28 endo-PGs through MEGA6. For multiple sequence alignment and phylogenetic analysis, sequences included were derived from *A. nidulans* (AnEPG, GeneBank accession number AN8327.2), *Aspergillus niger* (pgaII, GenBank accession no. CAA41694) [30], *Aspergillus aculeatus* (AaEPG, GenBank accession no. AC23565) [15], *Colletotrichum lupini var. setosum* (CluPG1, GenBank accession no ABL01533) [31], *Achaetomium* sp. Xz8 (PG I, GenBank accession no. AGR51994) [22], *Thielavia arenaria* XZ7 (endo-PG7fn, GenBank accession no. AIZ95162) [24], *Penicillium oxalicum* CZ1028 (EPG4, GenBank accession no. APZ75903.1) [17], *Aspergillus mulundensis* (hypothetical protein, GenBank accession no. XP_026600476.1), *Aspergillus calidoustus* (putative glycoside hydrolase family 28, GenBank accession no. CEN62944.1), *Penicillium antarcticum* (hypothetical protein, GenBank accession no. OQD89725.1). The signal peptide was predicted using the SignalP 4.1 Server (http://www.cbs.dtu.dk/services/SignalP/). The potential N- and O-glycosylation sites were predicted online (http://www.cbs.dtu.dk/services/NetNGlyc/, http://www.cbs.dtu.dk/services/NetOGlyc/).

### 3.11. Structure Modelling of AnEPG

Modelling was performed by using I-TASSER [32], and three enzymes were used as templates, including endo-polygalacturonase from *Aspergillus aculeatus* (PDB ID: 1IA5 and 1IB4) [33], endo-polygalacturonase II from *Aspergillus niger* (PDB ID: 1CZF) [30], and endo-polygalacturonase from *Colleotrichum lupini* (PDB ID: 2IQ7) [31].

## 4. Conclusions

This newly characterized endo-polygalacturonases from *A. nidulans* exhibited moderate activity under acidic conditions and good stability over a wide range of pH and below 40 °C. This pectinase has great potential to be used in the fields where acidic endo-polygalacturonases are required. 

In summary, multiple sequence alignment and phylogenetic analysis of AnEPG and homologous GH 28 endo-PGs suggested that AnEPG is a new endo-polygalacturonase. AnEPG was overexpressed in *Pichia pastoris* and characterized in detail. AnEPG showed the highest activity at pH 4.0 and 60 °C. It was very stable between pH 2.0 and pH 12.0. AnEPG exhibited high stability below 40 °C and was unstable above 50 °C. The apparent *K_m_* and *k_cat_* values of AnEPG against PGA (polygalacturonic acid) were 8.3 mg/mL and 5640 μmol/min/mg respectively. Ba^2+^ and Ni^2+^ showed some stimulatory effects on AnEPG, while AnEPG was inhibited by Ca^2+^, Cu^2+^, and Mn^2+^. The structural characteristics of endo-polygalacturonases were demonstrated by structure modelling of AnEPG. This pectinase could be potentially used in the beverage industry and/or other fields requiring acidic endo-polygalacturonases.

## Figures and Tables

**Figure 1 ijms-21-02100-f001:**
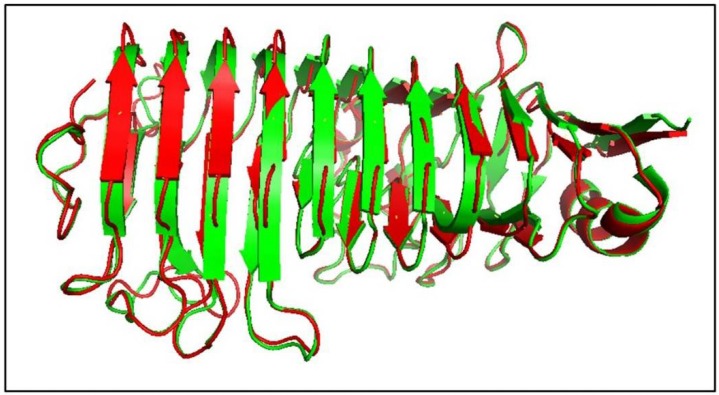
Structure modelling of AnEPG. Superimposition of structures of modelled AnEPG (red) and AaEPG (PDB ID: 1IA5) (green) [33]. The amino acid sequence identity between AnEPG and AaEPG was 57.8%.

**Figure 2 ijms-21-02100-f002:**
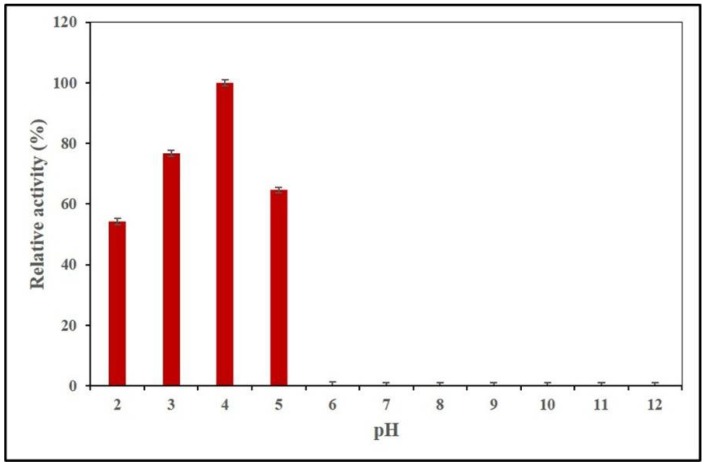
Effects of pH on AnEPG activity.

**Figure 3 ijms-21-02100-f003:**
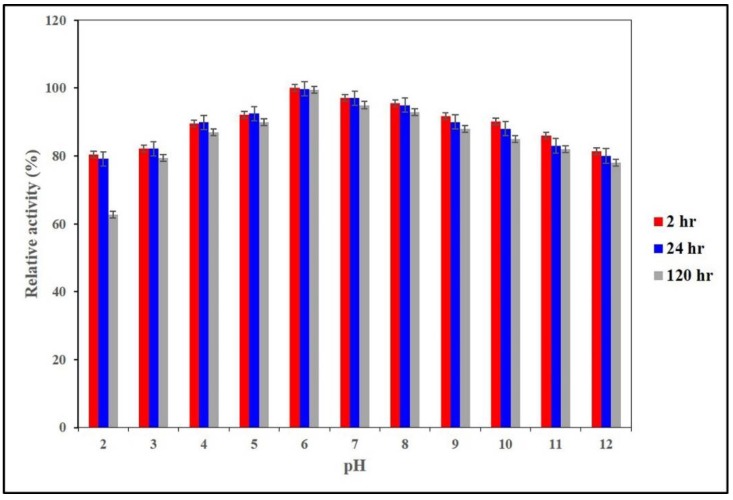
pH stability of AnEPG.

**Figure 4 ijms-21-02100-f004:**
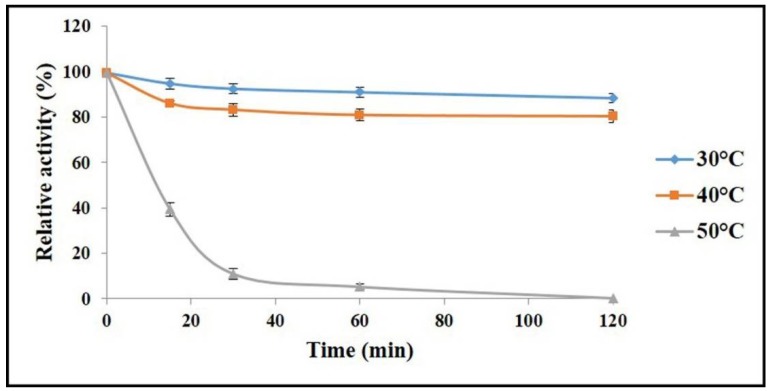
Thermal stability of AnEPG.

**Figure 5 ijms-21-02100-f005:**
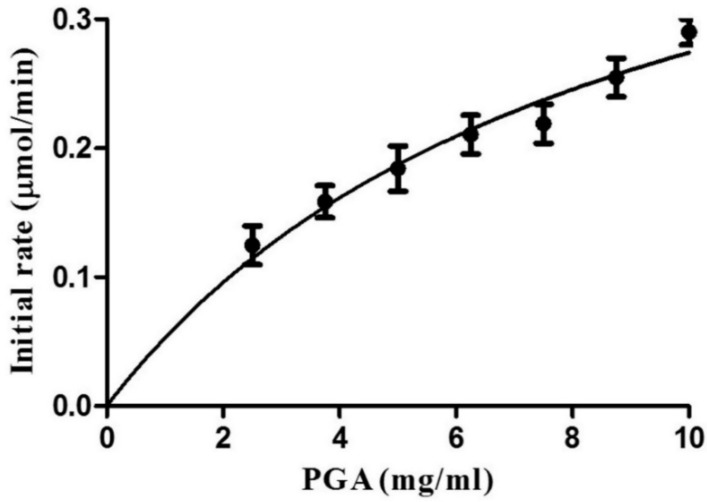
Effects of PGA (polygalacturonic acid) concentration on AnEPG activity.

**Figure 6 ijms-21-02100-f006:**
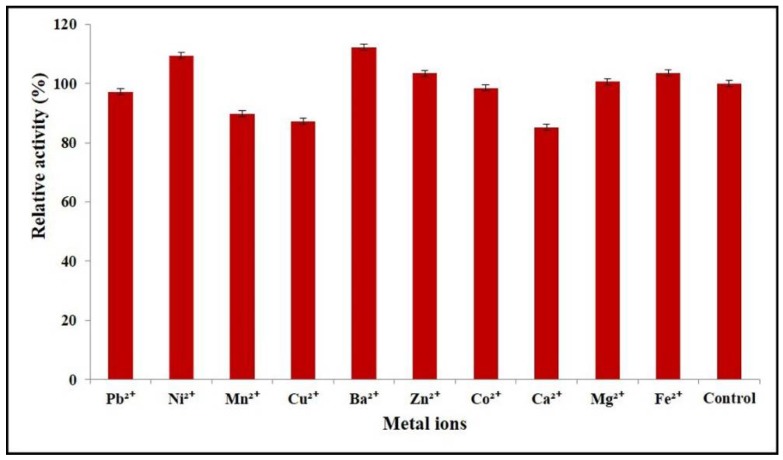
Effects of divalent metal ions on AnEPG activity.

**Figure 7 ijms-21-02100-f007:**
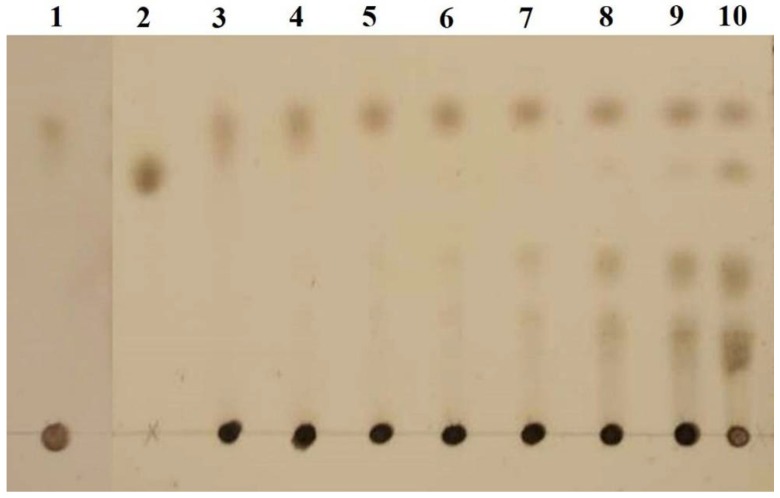
TLC (thin-layer chromatography) analysis of the hydrolysis of PGA by AnEPG. Hydrolysis of PGA was completed in 50 mM B & R (Britton and Robinson) buffer (pH 4.0) at 37 °C for 5 min, 10 min, 20 min, 30 min, 1 h, 2 h, 3 h, and 24 h, respectively. The hydrolysis products were analyzed by TLC on the silica gel plate. Lane 1: the control in the absence of enzyme AnEPG; Lane 2: galacturonate; Lanes 3–10: enzymatic hydrolysis products of PGA by AnEPG after being incubated for 5 min, 10 min, 20 min, 30 min, 1 h, 2 h, 3 h, and 24 h, respectively.

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
