# Peer review of "Overexpression and Biochemical Characterization of an Endo-α-1,4-polygalacturonase from Aspergillus nidulans in Pichia pastoris"

_ijms, 2020, doi:10.3390/ijms21062100_

Round 1

Reviewer 1 Report

This is an interesting paper detailing the characterization of an endo-polygalacturonase ferm A. nidulans. The manuscript is well-organized, although the language quality is slightly below the required standard. It should be publishable after a few changes in presentation and an increase in the rigor of the claims sourced from the sequence alignments. I would also request the authors to imporve the characterization of the model as detailed below.

COMMENTS:

lines 71-73:authors state "Multiple sequence alignment and phylogenetic analysis of
AnEPG [...] and other GH 28 endo-polygalacturonases implied that AnEPG is a new endo-
polygalacturonase" but do not cite any reference.

Fig 1.A) is almost unreadable. After increasing its size by 175% I managed to analyse it but it is not at all evident why this figure is grounds for the authors' observation (line 88) that AnEPG is "a new endo-polygalacturonase": apart from the very beginning of the N-terminal (where similarity seems to be relatively low among all sequences) high levels of homology are present throughout the alignment, no large gaps/insertions are present, all dissulfide bonds seem to be conserved, as are the catalytic aminoacids. The new sequence furthermore clusters quite nicely with other sequences with high bootstrap values. Authors should either tone down their claim of novelty or subtantially detail the basis for their claims.

line 96: how do you know that a dissulfide bond is formed between C369 and C378? This seems very speculative to me.

line 102: an RMSD of 7.6 angstrom is astonishingly high and definitely NOT a sign of a good model. However, he red and green structures in Fig.2 seems relatively similar, and therefore I cannot understand why such a high RMSD was found. Did the authors fail to remove the 40 initial N-terminal aminoacids (which are absent from the template) before computing the RMSD? 

Authors do not make use of the model, unfortunately. A comparison of the binding cavity with that of the template could be helpful, e.g. by showing possible differences in the aminoacids lining it, that might be helpful in rationalizing the differences in Km between template and target enzyme.

line 163 seems to contain a mistake ", its apparent K m value is significantly lower". I think the authors men ", its apparent K m value is significantly HIGHER" instead.

lines 172-175 the activity changes upon addition of metal ions seem to be residual. this should be discussed.

lines 176-193 are not easy to follow. These data should be presented as a table instead, preferably including the numerical values of the effect of those ions on the enzymes discussed.

Reviewer 2 Report

I found that the manuscript presents very interesting work and I recommend it for publication in International Journal of Molecular Sciences. I have to add, that the ms is written in good English. It is easy to read and understand the content. I found only one typos, it is on page 2 in line 44 where instead of 'methy' should be 'methyl'.

Author Response

Reviewer 2

Comments and Suggestions for Authors

I found that the manuscript presents very interesting work and I recommend it for publication in International Journal of Molecular Sciences. I have to add, that the ms is written in good English. It is easy to read and understand the content. I found only one typos, it is on page 2 in line 44 where instead of 'methy' should be 'methyl'.

Reply: As suggested by the reviewer, that was revised. We appreciate the reviewer’s comment on this.

Round 2

Reviewer 1 Report

I am mostly satisifed with the responses but I still feel that the modeling exercise has not been used to the utmost: authors should include the active site description (and respective comparison to the template active site) in the manuscript. I also believe that the RMSD comutation suffers from some hidded error, and I would like to compare the model coordinates to the template myself. In any case, those coordinates should be provided as supporting information.
